evolution, genetics

ageing, senescence, life-history evolution, developmental theory of ageing, lifespan

**Authors for correspondence:**
Martin I. Lind
e-mail: martin.lind@ebc.uu.se
Alexei A. Maklakov
e-mail: a.maklakov@uea.ac.uk

†These authors contributed equally to this paper.

# Cost-free lifespan extension via optimization of gene expression in adulthood aligns with the developmental theory of ageing

Martin I. Lind[1,†], Hanne Carlsson[2,†], Elizabeth M. L. Duxbury[2], Edward Ivimey-Cook[2] and Alexei A. Maklakov[2]

[1]Animal Ecology, Department of Ecology and Genetics, Uppsala University, Uppsala, SE-75236, Sweden
[2]School of Biological Sciences, University of East Anglia, Norwich NR4 7TJ, UK

MIL, 0000-0001-5602-1933; EMLD, 0000-0002-5733-3645; EI-C, 0000-0003-4910-0443; AAM, 0000-0002-5809-1203

Ageing evolves because the force of selection on traits declines with age but the proximate causes of ageing are incompletely understood. The 'disposable soma' theory of ageing (DST) upholds that competitive resource allocation between reproduction and somatic maintenance underpins the evolution of ageing and lifespan. In contrast, the developmental theory of ageing (DTA) suggests that organismal senescence is caused by suboptimal gene expression in adulthood. While the DST predicts the trade-off between reproduction and lifespan, the DTA predicts that age-specific optimization of gene expression can increase lifespan without reproduction costs. Here we investigated the consequences for lifespan, reproduction, egg size and individual fitness of early-life, adulthood and post-reproductive onset of RNAi knockdown of five 'longevity' genes involved in key biological processes in *Caenorhabditis elegans*. Downregulation of these genes in adulthood and/or during post-reproductive period increases lifespan, while we found limited evidence for a link between impaired reproduction and extended lifespan. Our findings demonstrate that suboptimal gene expression in adulthood often contributes to reduced lifespan directly rather than through competitive resource allocation between reproduction and somatic maintenance. Therefore, age-specific optimization of gene expression in evolutionarily conserved signalling pathways that regulate organismal life histories can increase lifespan without fitness costs.

## 1. Introduction

The force of natural selection is maximized during pre-reproductive development but declines after sexual maturation with advancing age [1–4]. Therefore, mutations that have neutral or positive fitness effects early in life but negative fitness effects late in life can accumulate (mutation accumulation theory [1]) or be selected for (antagonistic pleiotropy theory [2]) in the population and lead to the evolution of ageing [2,3]. Both mutation accumulation and antagonistic pleiotropy theories have received empirical support and probably contribute to the evolution and expression of ageing (reviewed in [5,6]). While these ultimate population genetic theories of ageing are broadly accepted, the proximate routes that lead to ageing are still incompletely understood and subject to vigorous debate [6–10]. The discovery of evolutionarily conserved molecular signalling pathways that regulate life-history traits, such as development, growth, reproduction and lifespan showed that ageing is malleable, and sometimes can be modified by modulating the expression of a single gene that influences a large array of downstream physiological processes

[6,11,12]. In line with the antagonistic pleiotropy theory developed by Williams [2], some studies show that the organismal life history can be modified from the one that focuses on early reproduction to the one the focuses on survival [6,9,13–15]. However, which proximate processes contribute to the observed effects remains unclear [6–9,13].

One proximate physiological account of the antagonistic pleiotropy theory, the 'disposable soma' theory of ageing (DST), postulates that ageing and lifespan evolve as a result of optimized resource allocation between somatic maintenance and reproduction with the aim of maximizing reproductive output [16,17]. This theory predicts that increased investment in soma will increase survival at the cost of reduced reproduction, and vice versa, since they are assumed to compete for the same pool of resources. Indeed, there is corroborating evidence from laboratory (reviewed in [18]) and field (reviewed in [19]) studies suggesting that there is a link between increased reproduction and reduced lifespan. Nevertheless, the predominance of this theory has been increasingly challenged in recent years [6,9,10,20–26]. Studies in different model organisms have suggested that increased longevity and reduced reproduction can be uncoupled, thereby questioning the key role of resource allocation trade-offs in ageing (reviewed in [6,7,9,22]).

Nevertheless, Williams himself proposed a different mechanism underlying antagonistic pleiotropy, by suggesting that the declining force of selection with age can result in suboptimal levels of gene expression in late life [2]. Because selection is strongest during development and declines after the onset of reproduction [3], selection can never fully 'optimize' age-specific gene expression resulting in ageing via the action of otherwise beneficial genes. More recently, several authors further developed these ideas focusing on the role of suboptimal gene expression in adulthood in the evolution of ageing [9,10,25,27]. This developmental theory of ageing (DTA) maintains that the decline in selection gradients with age results in suboptimal regulation of gene expression in adulthood, leading to cellular and organismal senescence [9,25,27].

There is an important distinction between these two physiological explanations of how antagonistically pleiotropic alleles work [9]. The DST rests on the competitive allocation of resources between the soma and the germline resulting in imperfect repair of cellular damage; this theory predicts that genetic and environmental manipulations that increase allocation to the disposable soma (hence lifespan) result in reduced allocation to the immortal germline (hence reproduction) [9]. The DTA instead focuses on imperfect age-specificity of gene expression and predicts that optimizing gene expression in adulthood can improve the soma as well as the germline. Increased understanding of the evolutionarily conserved molecular pathways [11] that control many different aspects of organismal life cycle allows direct testing of these two explanations. Since the DTA is based on the assumption that gene function affects fitness differently across the life course of the organism, perhaps the most straightforward way to test it is to modify the gene expression at different stages across the life course and assess the effects on fitness-related traits and on individual fitness.

Here we tested these predictions directly by modifying the age-specific expression of five well-described 'longevity' genes in *Caenorhabditis elegans* nematode worms that play key roles in different physiological processes: nutrient-sensing signalling via insulin/IGF-1 (*age-1*) [28,29] and target-of-rapamycin (*raga-1*) [30,31] pathways, global protein synthesis (*ifg-1*) [32], global

protein synthesis in somatic cells (*ife-2*) [33,34] and mitochondrial respiration (*nuo-6*) [35]. The *age-1* gene encodes the phosphatidylinositol 3-kinase (PI3 K) catalytic subunit homologue, which is involved in the kinase-phosphorylation cascade that downregulates the DAF-16/FOXO transcription factor [29]. Loss-of-function mutations in *age-1* increase lifespan [28,29] but reduce early-life reproduction and fitness [15,36,37]. The *raga-1* gene encodes the *C. elegans* orthologue of GTPase RagA, which is the amino acid-sensing activator of the target-of-rapamycin complex 1 (TORC1) signal transduction pathway [30] that governs cell growth and shapes lifespan [38]. Loss-of-function *raga-1* mutants have a longer lifespan and slower behavioural decline with age [39]. The *ifg-1* gene encodes the *C. elegans* orthologue of the scaffold protein eIF4G, a part of the eIF4F complex, which mediates mRNA translation. Inhibition of *ifg-1* increases lifespan but reduces fecundity and slows down growth [32]. The *ife-2* gene encodes a eukaryotic translation initiation factor eIF4E, which is a regulator of protein synthesis and is most abundant in the somatic cells in *C. elegans*. Under standard temperature (20°C), disruption of *ife-2* via mutation or lifelong RNA interference (RNAi) increases survival without negative effects on brood size [34,40], and it is suggested that the lifespan extension is conferred specifically via reduction of protein synthesis in the soma [41]. The *nuo-6* gene encodes a subunit of mitochondrial complex I in the mitochondrial respiratory chain, and lifelong *nuo-6* RNAi reduces growth and fertility but increases longevity [35].

Our approach was to use age-specific RNAi to downregulate the expression of these genes starting at three different stages across the life course of *C. elegans*: (i) newly laid egg (lifelong treatment), (ii) sexual maturity (adulthood treatment) and (iii) the end of self-fertilized reproduction (post-reproductive treatment). This approach allowed us to assess the fitness consequences of lifelong and adulthood-only downregulation of the target genes, as well as the effects of post-reproductive downregulation on survival. The latter effect is particularly interesting in this regard, because it allows us to test whether age-specific optimization of gene function can extend lifespan in the absence of the cost of reproduction. If post-reproductive downregulation of gene expression can indeed increase survival, it would be a proof-of-principle that gene expression in late life is not optimized for long life. Nevertheless, an even more crucial test was whether we could modify different physiological functions during the reproductive period of adulthood to increase lifespan without hampering reproduction. We investigated age-specific RNAi effects on survival, age-specific reproduction and egg size (as a measure of parental investment and a proxy for offspring quality); we then used these data to determine lifetime reproductive success (LRS) and rate-sensitive individual fitness ($\lambda_{ind}$). Individual fitness integrates the reproduction across age classes, with emphasis on early reproduction and thus fast development, into a single metric. It is therefore the most appropriate fitness measure when timing is important for fitness [42].

## 2. Methods

### (a) Strains

*Caenorhabditis elegans* nematodes of strain Bristol N2 wild-type, obtained from Caenorhabditis Genetics Center, were used in all assays. Populations were recovered from frozen stocks and

bleached before the start of the experiment. Standard NGM agar plates [43] were used to grow the nematode populations and antibiotics (100 μg ml$^{-1}$ ampicillin and 100 μg ml$^{-1}$ streptomycin) and a fungicide (10 μg ml$^{-1}$ nystatin) was added to the agar to avoid infections [44]. Up until the start of the experiment, nematode populations were fed antibiotic-resistant *Escherichia coli* OP50-1 (pUC4 K), gifted by J. Ewbank at the Centre d'Immunologie de Marseille-Luminy, France. During recovery from freezing and throughout the experiment, the worms were retained in climate chambers maintaining 20°C and 60% relative humidity.

In order to induce RNAi knockdown, the nematodes were fed *E. coli* of the strain HT115 (DE3) containing a Timmons and Fire feeding vector L4440 modified to express the dsRNA of the gene of interest. The genes targeted by RNAi were *age-1*(B0334.8), *raga-1* (T24F1.1), *nuo-6* (W01A8.4), *ife-2* (R04A9.4) and *ifg-1* (M110.4). These strains were provided by Source Bioscience and Julie Ahringer. In addition to the RNAi knockdown bacteria, a control strain of HT115 was used, carrying an empty L4440 feeding vector.

Cultures of the RNAi clones were grown in LB medium supplemented with ampicillin (50 μg ml$^{-1}$) before seeding onto 35 mm standard NGM plates with the addition of IPTG (1 mM) and ampicillin (50 μg ml$^{-1}$). After seeding, the bacteria were allowed to grow overnight in 20°C to induce expression of RNAi, before worms were placed on the plates.

## (b) Experimental set-up

Age-synchronized eggs were collected from OP50-1 (pUC4 K) fed unmated hermaphrodites on adult day 2. These eggs were placed either on RNAi seeded plates and maintained on RNAi throughout life (*lifelong exposure*), on an empty vector from egg to late L4 stage and then onto RNAi (*adulthood exposure*), on an empty vector from egg to day 6 of adulthood and then on RNAi (*post-reproductive exposure*), or maintained throughout life on empty vector plates (*control*). For every gene knockdown, assays were performed in two blocks. The scoring was achieved by a blinded observer, with agar plates of the different treatments handled in a randomized order.

## (c) Reproductive assays

To gather daily reproductive output per worm, unmated hermaphrodites were reared on individual plates from late L4 stage until reproduction ceased. The worms were moved onto new plates every 24 h. Eggs laid on the plates were allowed to hatch and develop during 2 days, when the total amount of worms on the plates were counted. For each block, 15 replicate worms were set up for each gene and treatment combination, giving 30 worms per gene and treatment (120 worms per gene in total, across all four treatments), except for the second block of *ife-2* and *raga-1*, where more worms were set up in order to compensate for lost worms in the first block.

## (d) Lifespan assays

Lifespan assays were performed on unmated hermaphrodites from larval stage L4 until death. Worms were set up in groups of ten and transferred onto fresh plates daily while scoring survival. Death was defined as the absence of movement in response to touch. For each block, 50 replicate worms were set up for each gene and treatment combination, giving 100 worms per gene and treatment (400 worms per gene in total, across all four treatments), except for the second block of *ife-2* and *raga-1*, where more worms were set up in order to compensate for lost worms in the first block.

## (e) Egg size measurement

Egg photos were taken from the same plates as used in the lifespan assays. At adult day 2, the 10 lifespan worms were allowed to lay eggs for 2 h on a fresh agar plate, after which photos of 10 eggs were taken per plate of 10 worms. A microscope camera was used to attain the photos, which were later analysed in ImageJ (https://imagej.nih.gov/ij/) to score cross-sectional area in mm$^2$.

## (f) Validation of gene expression downregulation

We quantified the extent of downregulation of the five target genes following feeding RNAi, using quantitative reverse transcriptase-polymerase chain reaction (qRT-PCR).

First, we set up a separate batch of worms (from the same frozen population of N2) identically to those in the main experiment for the lifelong treatment (RNAi applied from egg stage onwards) and age-synchronized as before. There were six treatments in total: individuals were either fed RNAi targeting one of the five genes above, or empty vector control bacteria. In total, there were 90 worms per treatment, stored 10 per plate.

Worms were collected on day two of adulthood, to assay gene expression downregulation at peak reproduction. We first pooled worms into three groups of 30 worms, by picking onto unseeded plates and allowing worms to crawl around to remove surface bacteria and separate worms from their eggs as in [45]. Worms were then washed in M9 buffer three times following [46] then most of the M9 was removed and the worms were suspended in 500 μl of TRISure reagent (Bioline), flash frozen in liquid nitrogen and stored at −80°C. Pooling worms generated three biological replicates per RNAi treatment, each with 30 worms, to extract sufficient RNA for qRT-PCR [45]. Further method details are included in electronic supplementary material.

## (g) Statistical analyses

All analyses were performed separately for each gene. Before analysis, we excluded individuals from plates that were severely contaminated by infection. We also removed one infertile control individual (electronic supplementary material, table S1–S2). All statistical analyses were performed using the statistical software R 3.6.0.

Lifespan was analysed using Cox proportional hazard models implemented in the *coxme* package [47], with treatment (age at RNAi treatment exposure) as a fixed factor, and block and plate as random effects. Individuals dying of matricide (internal hatching of eggs) were censored. In addition to lifespan, we also analysed the age-specific mortality rate using the *BaSTA* package [48]. We first investigated age-specific mortality rates using Weibull, Gompertz and logistic functions, with either a simple, Makeham or bathtub shape. In addition, we also modelled a scenario without senescence (exponential model with a simple shape). All models were run as four independent simulations; each with 600 000 iterations, where the first 6000 iterations were discarded as burn in, and the model was sampled every 600 iterations. The models were compared using DIC and the best model was chosen for each gene.

The best mortality functions (lowest DIC) were the Gompertz model with a simple shape (for *age-1*, *ife-2*), the logistic model with a simple shape (for *ifg-1*, *raga-1*) and the Weibull model with a bathtub shape (for *nuo-6*). For the Gompertz model, the two beta parameters describe how mortality changes with age; $b_0$ is the age-independent, baseline mortality rate and $b_1$ describes the exponential increase of mortality with age. For logistic models, the additional beta parameter $b_2$ is the asymptote of the curve, describing the deceleration of mortality at old ages. Finally, the Weibull model has two beta parameters describing the mortality function, but additionally the two alpha parameters ($a_0$ and $a_1$) describe an exponential decline early in life, and the constant (c) is the lowest point of the mortality function. It should be noted that the beta parameters are not comparable across models, since in Gompertz models, mortality

increases exponentially with time, in Weibull models, the increases over time follow a power function, and in logistic models, the mortality first increases and then decreases in late life [49]. For statistical comparisons of treatment effects, we used the Kullback–Leibler divergence calibration (KLDC). Values close to 0.5 imply minor differences between distributions, while values closer to 1 imply major differences. Following established practice [50,51], we consider values greater than 0.8 imply substantially different distributions.

Age-specific reproduction was analysed using generalized linear mixed-effect models. We used the first 3 days of reproduction, since reproduction ceased on day 4. We treated treatment, age (days) and age$^2$ as crossed fixed factors, and fitted block as well as individual as random effects (to control for repeated measures). The addition of age$^2$ to the models was to account for curvature. Since reproduction data was often overdispersed, we fitted up to seven different model implementations. First, we fitted the models using a Poisson distribution in the *lme4* package [52]. Second, we also included an observation-level random effect in the model, to control for possible overdispersion. Third, we fitted a model with a Conway–Maxwell–Poisson (CMP) distribution using the *glmmTMB* package, where the mean and variance are allowed to vary independently, and is well suited to deal with both over- and under-dispersed data [53]. We tested for zero inflation and over/under-dispersion using the *DHARMa* package [54]. If significant zero inflation was detected, it was modelled using zero-inflated CMP models (ZICMP). If significant dispersion was detected (if necessary even after modelling zero inflation), we also included CMP models with different dispersion models, where dispersion was allowed to vary with the level of the covariate (age and age$^2$). The models were then compared using AIC and the model with lowest AIC selected. In one case (*nuo-6*), the final model was still under-dispersed, which makes the *p*-values obtained overly conservative and reduces our power to detect significant effects [55]. Significance of the final model was determined using type III Wald $\chi^2$ tests implemented in the *car* package [56].

Individual fitness ($\lambda_{\text{ind}}$) was calculated from the life-table of age-specific reproduction, by solving the Euler–Lotka equation using the lambda function in the *popbio* package [57]. The life-table was constructed using a common development time of two days without reproduction. Reproduction started during day three (first day of adulthood), where worms matured during the day. Thus, fast-developing worms would mature earlier during this day and consequently have more reproduction at this day. Since $\lambda_{\text{ind}}$ is a rate-sensitive measure, fast development, resulting in increased early reproduction, would translate into higher values of $\lambda_{\text{ind}}$. We then analysed $\lambda_{\text{ind}}$ in linear mixed-effect models using the *lme4* package, with treatment as a fixed factor and block as a random effect. Significance was determined using type III Wald $\chi^2$ tests implemented in the *car* package.

Lifetime reproductive success (LRS) was scored as the total number of offspring per individual and was analysed using generalized linear mixed-effect models using Poisson or CMP distribution, as described above for age-specific reproduction. CMP models accounting for zero inflation were fitted only if significant zero inflation was detected using *DHARMa*. We fitted treatment as a fixed factor and block as a random effect. Significance of the final model was determined using type III Wald $\chi^2$ tests implemented in the *car* package.

Individual fitness and LRS were also analysed by bootstrapping, using the *dabestr* package [58], and the 95% confidence intervals are graphically presented by the package in figure 2 and electronic supplementary material, S1.

Egg size was analysed in linear mixed-effect models using the *lme4* package, with Treatment as a fixed effect, and block and plate as random effects. Treatments without any eggs produced were not included (lifelong exposure to *ifg-1*).

Significance was determined using type III Wald $\chi^2$ tests implemented in the *car* package.

To calculate relative gene expression, we determined ΔCt as the difference between the qRT-PCR cycle thresholds (Ct values) of the target gene of interest and the reference gene, for each sample. The arithmetic mean of the Ct values for the two technical replicates per gene, per sample was used in delta Ct calculations. Statistical analyses were performed on ΔCt as in [45,59], using a linear model with Gaussian error structure, to determine the effect of RNAi treatment (RNAi versus empty vector controls) on relative gene expression for all target genes. RNAi treatment, gene and their interaction were fitted as categorical factors. The Shapiro–Wilk normality test and visual inspection of quantile-quantile plots confirmed that ΔCt values satisfied the normality assumption of the linear models (electronic supplementary material, table S20).

The coefficient of variation (CV, %) in ΔCt between biological replicates for each RNAi treatment, and in Ct values between technical replicates per gene, per sample, was calculated as the standard deviation divided by the mean for each comparison as in [45,60], to determine biological variation in relative gene expression between pools of 30 worms and repeatability of the qPCR results, respectively. To quantify fold change in gene expression ($2^{-\Delta\Delta CT}$), we calculated the difference in the relative levels of mRNA for the target gene of interest compared to the reference gene (ΔCt) between untreated controls and RNAi treated samples, using mean ΔCt from the three biological replicates per RNAi treatment as in [61].

## 3. Results

Timing of RNAi treatment had profound effects on survival, mortality rate, age-specific and lifetime reproduction, egg size and fitness (figures 1 and 2, electronic supplementary material, figure S1–6, tables 1 and 2). Nevertheless, there was little evidence for a link between increased lifespan and reduced fitness. Downregulation of *age-1* across all life stages improved longevity and decreased age-specific mortality rate (figure 1 and electronic supplementary material, S1 and tables S3–S5), but the effect became progressively weaker with increasing age of onset of the RNAi treatment; however, we found no indication that *age-1* RNAi negatively affected LRS, egg size or individual fitness ($\lambda_{\text{ind}}$) (figures 1 and 2; electronic supplementary material, figure S6; table 1; electronic supplementary material, tables S14–S18). Interestingly, the effect of TORC1 downregulation via *raga-1* RNAi on traits was quite different: lifelong RNAi did not have any positive effect, while adulthood-only and post-reproductive treatments slightly improved survival but did not affect LRS, egg size or individual fitness (figures 1 and 2; electronic supplementary material, figure S6; table 1; electronic supplementary material, tables S3, S14–S18). The survival benefit of *raga-1* downregulation was caused by a changed late-life asymptote in mortality rate, evident in all treatments (electronic supplementary material, figure S2 and tables S6–S7). These results suggest that the two major nutrient-sensing molecular signalling pathways, IIS and TOR, have very different effects on vital life-history traits.

Age-specific downregulation of *nuo-6* showed a perfect negative correlation between survival and reproduction. Similar to the results with *age-1*, *nuo-6* RNAi increased survival and the effect became weaker with increased age of onset of RNAi treatment (figure 1; table 2; electronic supplementary material, table S3). This was mirrored in the modulation of both

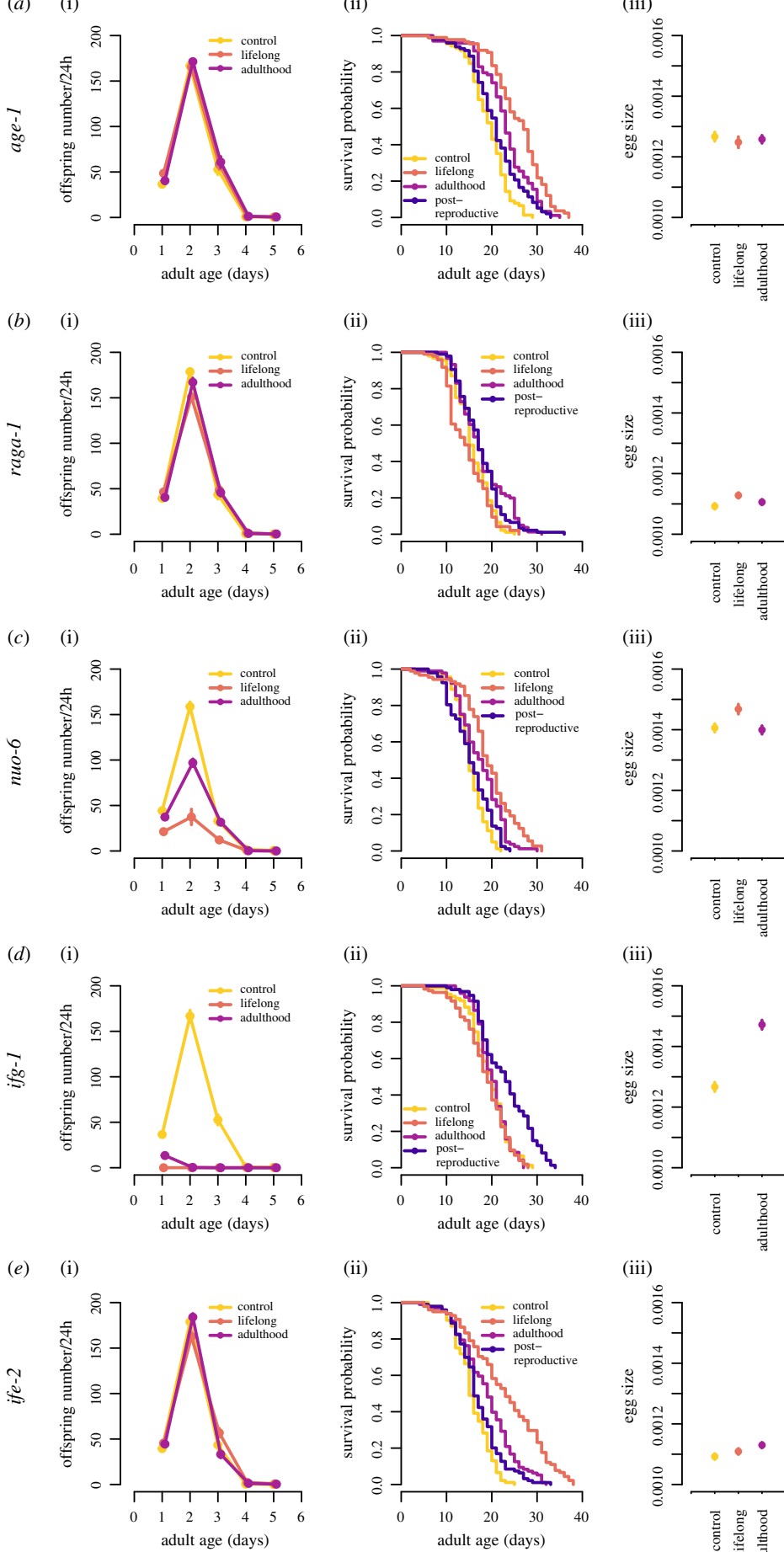

**Figure 1.** Age-specific effects of 'longevity' genes on traits. Age-specific reproduction, lifespan and egg size presented for each gene and separated by treatment group. Colours indicate control (yellow), lifelong RNAi treatment (orange), RNAi during adulthood only (purple) and post-reproductive RNAi (blue). For age-specific reproduction and egg size, symbols represent mean ± s.e.

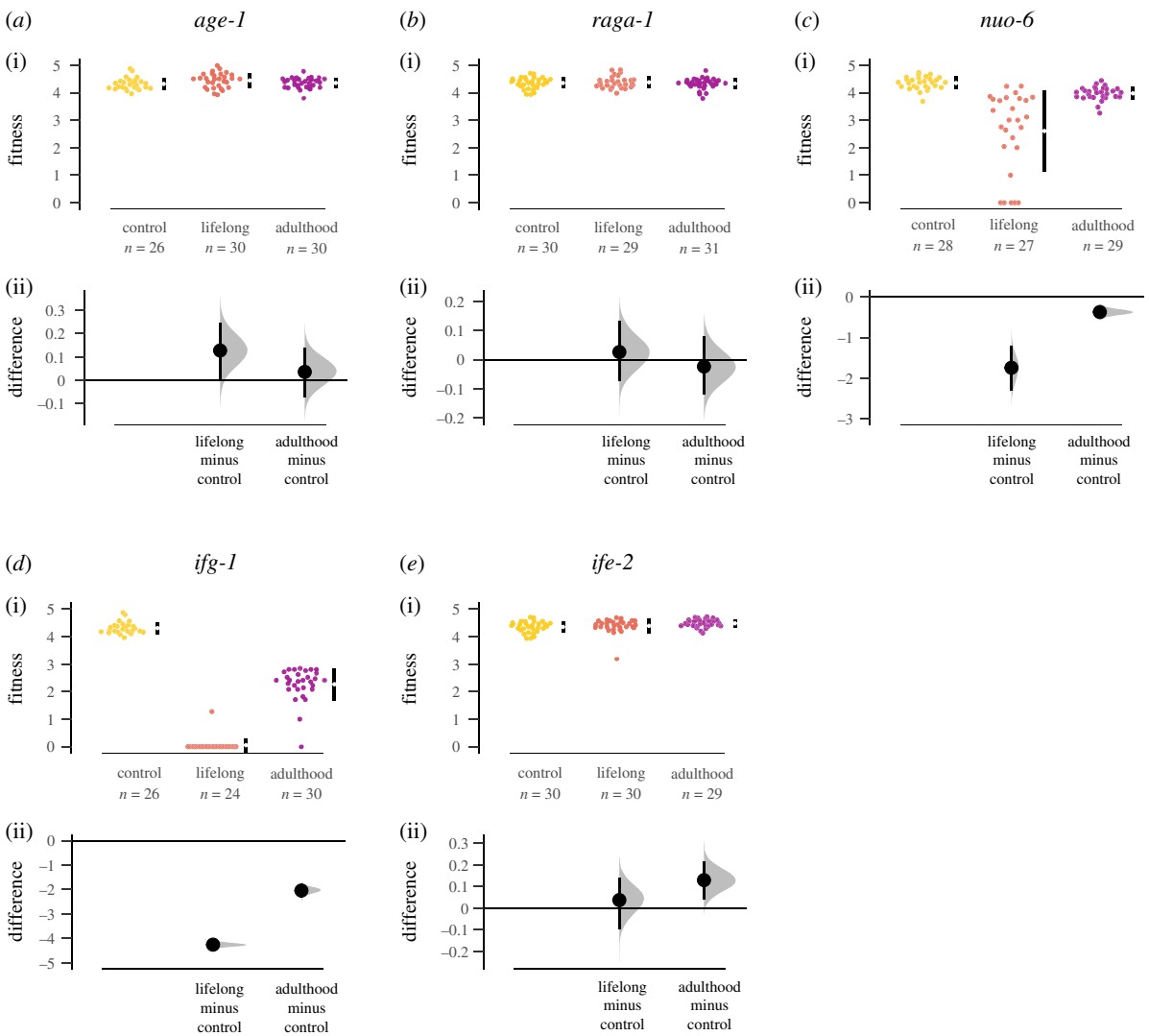

**Figure 2.** Age-specific effect of 'longevity' genes in fitness. Individual fitness ($\lambda_{ind}$) calculated from age-specific fecundity data, separated by gene and treatment group: control (yellow), lifelong RNAi treatment (orange) and RNAi during adulthood only (purple). Top panels show raw data, with the mean ± 95% CI indicated by black bars at each group. Bottom panels show estimation plots, where RNAi treatments are compared to the control, with a graded sampling distribution of bootstrapped values and the bootstrapped 95% CI.

age-independent and age-specific mortality rate (electronic supplementary material, figure S3 and tables S8–S9). Contrary to the effect of *age-1* RNAi, however, improved survival was mirrored by negative effects on LRS and fitness, while egg size was improved in the lifelong treatment (figures 1 and 2; electronic supplementary material, figure S6; tables 1 and 2; electronic supplementary material, tables S16–S18).

Downregulation of *ifg-1* predictably abolished reproduction in the lifelong treatment, and severely reduced it when started in adulthood, while the few eggs produced in the adulthood-only treatment were quite large (figure 1; electronic supplementary material, figure S6; table 1; electronic supplementary material, table S14–S18). Interestingly, there was no effect of reduced reproduction on survival (figure 1; table 2; electronic supplementary material, table S3). Perhaps even more remarkable was the positive effect of post-reproductive *ifg-1* RNAi on survival (figure 1 and table 2), where it especially stands out with its low late-life asymptote in mortality rate (electronic supplementary material, figure S4 and tables S10–S11). These results support the notion that superfluous protein synthesis in late life reduces longevity in *C. elegans*.

Age-specific downregulation of *ife-2* increased survival across all treatments with the effect becoming progressively

weaker with the later age of RNAi onset (figure 1 and table 1; electronic supplementary material, table S3), similar to the results with *age-1* and *nuo-6*. This was also mirrored by strong treatment effects in both age-independent and age-specific mortality rate (electronic supplementary material, figure S5 and tables S12–S13). Interestingly, there were no negative effects on LRS (electronic supplementary material, figure S6; table 1; electronic supplementary material, table S15), while adulthood-only RNAi actually increased egg size (figure 1; tables 1 and 2; electronic supplementary material, table S16). Thus, adulthood-only *ife-2* RNAi simultaneously improved survival and investment in offspring; moreover, within-model contrasts suggest adulthood-only *ife-2* RNAi had higher individual fitness than control animals (electronic supplementary material, table S14), although there was no overall significant effect across all three treatments (table 1). Nevertheless, bootstrapping analyses, which does not depend on a specified distribution of the data, suggests that adulthood-only *ife-2* RNAi does increase rate-sensitive fitness (figure 2).

We confirmed, using qRT-PCR analysis, that feeding nematodes bacteria that express double-stranded RNA for the target gene of interest significantly reduced mRNA

**Table 1.** The overall effect of downregulating *age-1*, *raga-1*, *nuo-6*, *ifg-1* or *ife-2* on lifespan, fitness ($\lambda_{ind}$), lifetime reproductive success (LRS) and egg size.

| gene | factor | $\chi^2$ | d.f. | p | $\chi^2$ | d.f. | p |
|---|---|---|---|---|---|---|---|
| | | lifespan | | | fitness ($\lambda_{ind}$) | | |
| *age-1* | treatment | 137.97 | 3 | <0.001 | 4.838 | 2 | 0.089 |
| *raga-1* | treatment | 45.681 | 3 | <0.001 | 0.986 | 2 | 0.611 |
| *nuo-6* | treatment | 206.72 | 3 | <0.001 | 66.352 | 2 | <0.001 |
| *ifg-1* | treatment | 108.43 | 3 | <0.001 | 541.13 | 2 | <0.001 |
| *ife-2* | treatment | 110.89 | 3 | <0.001 | 5.485 | 2 | 0.064 |
| | | LRS | | | egg size | | |
| *age-1* | treatment | 1.484 | 2 | 0.476 | 0.26 | 2 | 0.878 |
| *raga-1* | treatment | 3.024 | 2 | 0.221 | 3.30 | 2 | 0.192 |
| *nuo-6* | treatment | 89.389 | 2 | <0.001 | 9.40 | 2 | 0.009 |
| *ifg-1* | treatment | 1273.000 | 2 | <0.001 | 151.65 | 2 | <0.001 |
| *ife-2* | treatment | 0.707 | 2 | 0.702 | 6.45 | 2 | 0.040 |

levels and downregulated gene expression for all targeted genes (electronic supplementary material, table S21). The extent of gene expression downregulation did not depend on the specific gene target of the RNAi (no RNAi treatment x gene interaction), although the mean downregulation varied between 26% and 60% (based on gene expression fold change, $2^{-\Delta\Delta CT}$; electronic supplementary material, figure S7).

Technical replicates were highly repeatable for all samples (*actin-3* reference gene, CV < 0.6%; target genes, CV < 1% for all except two samples that were CV < 2.5%), confirming the repeatability of the qRT-PCR assay. Biological replicates showed considerably more variation in relative gene expression within each RNAi treatment by gene combination (mean CV = 9%, max. CV = 21%; electronic supplementary material, figure S8), but within the range expected for *C. elegans* based on previous qRT-PCR expression analyses for different genes as in [45,60].

## 4. Discussion

The two physiological theories that aim to explain the mechanistic basis of antagonistic pleiotropy have very different predictions. The DST proposes that improved somatic maintenance necessitates increased resource allocation, which will lead to reduced investment in growth and reproduction. Contrary to this, the DTA maintains that survival can be improved by optimizing age-specific gene expression without reproduction costs because gene expression is predicted to be optimized for development and early-life reproduction. The corollary of this argument is that optimizing gene expression during adulthood can increase individual fitness. The force of natural selection declines with age and does so very rapidly in small fast-reproducing organisms such as *C. elegans* [62]. This means that even very small positive effects on vital life-history traits early in life can be beneficial for individual fitness despite large fitness costs late in life [3,62]. This also implies that natural selection on regulating gene expression in late life is very weak in *C. elegans* and there is scope for experimental optimization of age-specific gene expression.

We found that four out of five 'longevity' genes that we tested showed poor correlation between the age-specific gene expression effects in survival, reproduction, egg size and individual fitness (figures 1 and 2; table 1). Only one of these genes—*nuo-6*—showed the pattern of a negative correlation between an increase in survival and reduced reproduction and fitness that is predicted under the DST. We note that such a correlation does not imply causation, and it is possible that upregulation of stress resistance pathways in *nuo*-6 does not depend on resource reallocation from reduced egg laying. The results for *nuo-6* support the previous work showing that rates of ageing are affected by mitochondrial function during development [63]. Nevertheless, our result also showed that adulthood-only reduction in mitochondrial respiration also could extend lifespan.

Three other genes that we tested (*age-1*, *raga-1* and *ife-2*) show the pattern that is partially consistent with the DST because late-life RNAi had a weaker effect on lifespan than early-life RNAi. Such a pattern is expected under DST because less cellular damage would accumulate if resource reallocation towards somatic maintenance would start at an earlier age. However, we did not find a corresponding reduction in fecundity; in other words, early downregulation of gene expression tended to increase lifespan relatively more in some genes but there was no cost to reproduction. The lack of negative effects on reproduction was not caused by the lack of power to detect them, because the treatment means were actually positive for most comparisons (figure 2; electronic supplementary material, figure S6). On the other hand, this pattern is similarly consistent with other ultimate and proximate theories of ageing, including the AP and the DTA. Indeed, using Williams's own abstract example, we can imagine an allele that improves a physiologically important process like bone calcification in development, which later in the life cycle causes increased calcification of arteries in an adult organism, and therefore reduces adult survival [2]. If reduction in survival occurs sufficiently late in the life cycle when natural selection is weak [3,4], then (i) such an allele can go to fixation and (ii) there will be little selection to modify its expression in adulthood [2]. However, the earlier we downregulate the expression of such an allele, the less damage to arteries will accumulate with age.

**Table 2.** Age-specific reproduction. The effect of RNAi treatment, age and age$^2$ on age-specific reproduction for each of the five genes. All final models were fitted using a Conway–Maxwell–Poisson (CMP) distribution (models with CMP distribution had lowest AIC for all genes, see electronic supplementary material, table S7) modelling dispersion to vary with age and age$^2$. It was not possible to model age-specific reproduction for *ifg-1*, since most treatment levels and ages lacked reproduction (see fitness and LRS instead). Right column: italics indicate significance at *p* < 0.05.

| gene | factor | $\chi^2$ | d.f. | *p* |
|------|--------|------|------|-----|
| *age-1* | intercept | 6.8 | 1 | *0.009* |
| | treatment | 12.5 | 2 | *0.002* |
| | age | 619.1 | 1 | *<0.001* |
| | age$^2$ | 455.8 | 1 | *<0.001* |
| | treatment × age | 9.6 | 2 | *0.008* |
| | treatment × age$^2$ | 6.1 | 2 | *0.048* |
| *raga-1* | intercept | 14.7 | 1 | *<0.001* |
| | treatment | 17.5 | 2 | *<0.001* |
| | age | 656.7 | 1 | *<0.001* |
| | age$^2$ | 514.6 | 1 | *<0.001* |
| | treatment × age | 16.0 | 2 | *<0.001* |
| | treatment × age$^2$ | 11.4 | 2 | *0.003* |
| *nuo-6* | intercept | 1.5 | 1 | 0.214 |
| | treatment | 8.8 | 2 | *0.013* |
| | age | 371.6 | 1 | *<0.001* |
| | age$^2$ | 318.8 | 1 | *<0.001* |
| | treatment × age | 19.6 | 2 | *<0.001* |
| | treatment × age$^2$ | 15.9 | 2 | *<0.001* |
| *ifg-1* | almost complete cross-separation, not possible to model | | | |
| *ife-2* | intercept | 18.0 | 1 | *<0.001* |
| | treatment | 19.0 | 2 | *<0.001* |
| | age | 762.5 | 1 | *<0.001* |
| | age$^2$ | 560.4 | 1 | *<0.001* |
| | treatment × age | 24.0 | 2 | *<0.001* |
| | treatment × age$^2$ | 23.8 | 2 | *<0.001* |

One interesting finding here was that in three of the longevity genes that we studied (*age-1*, *raga-1* and *ife-2*), we do not detect negative fitness consequences of lifelong downregulation of gene function. We must remember that feeding RNAi knockdowns only reduce gene expression rather than totally abolishing it. We used RNAi constructs from Ahringer library that were sequence verified prior to use and we recovered all of the classical longevity phenotypes in line with the previous research, so we do not have any reason to expect anything unusual with respect to our feeding RNAi protocol. We also validated the feeding RNAi approach and demonstrated that the expression of the five target genes was downregulated in two-day-old adults, the age of peak reproduction, following RNAi treatment from the egg stage.

Taken at face value, our results suggest that the expression of some of the genes can be reduced lifelong without strong negative effects on fecundity, at least in the benign environment. We do know that *age-1*, *raga-1* and *ife-2* mutants suffer from reduced fecundity [15,39,40]. This suggests that a mutation in one of these genes that increases lifespan and even perhaps late-life fecundity at the cost of early-life fecundity would be detrimental to fitness [15]. At the same time, because lifespan extension occurred in the late post-reproductive part of the life cycle, there would be little selection for a reduction of gene expression that mirrors the effects uncovered in our experiments.

Interestingly, adulthood-only RNAi knockdown of *ife-2* improved survival and egg size. These results suggest that superfluous protein synthesis in the somatic cells of adult worms promotes cellular senescence and reduces individual fitness through the effects on both parents and their offspring through egg size. More generally, we showed that adulthood-only, or even post-reproductive downregulation of important physiological functions can often improve survival without negative fitness effects. The results that we obtained in age-specific *age-1* and *ife-2* RNAi experiments suggest that adulthood-only knockdowns can increase fitness because they improve survival without negative effects on reproduction, and even a positive effect on egg size and individual fitness in the case of *ife-2*. Perhaps particularly intriguing is the fact that in four out of five cases, lifespan extension could be achieved via post-reproductive onset of RNAi treatment. While post-reproductive worms do not affect the allelic frequencies in the next generation, as there is no post-hatching parental care in this system, this latter result nevertheless suggests that late-life expression of these genes contributes to an earlier death.

While DTS and DTA make no explicit predictions about mortality rate, such analyses may still give further understanding of how lifespan extension is mediated. While different mortality rate models fitted different knockdowns, we found that lifespan extension was achieved by decreased age-specific mortality rate and in some cases also by lowered baseline mortality. However, for *raga-1*, downregulation by RNAi instead resulted in a lowered late-life deceleration in mortality rate, suggesting that the two major nutrient signalling pathways IIS (*age-1*) and TOR (*raga-1*) have different effects on life-history traits.

Overall, the results of this study are consistent with the hypothesis that selection optimizes gene expression in early-life, while post-maturation expression can be optimized further, as predicted by the developmental theory of ageing, a proximate physiological account of the more general antagonistic pleiotropy theory [9]. While the lack of selection on gene expression during the post-reproductive period is rather straightforward, one can question why the selection is so weak during the reproductive period of *C. elegans* life cycle. The answer probably lies in the biology of this species, which is characterized by a very rapid and strong (orders of magnitude) age-specific decline in selection gradients [62]. Indeed, the selection gradients on fecundity decline nearly exponentially with age in the laboratory [62], and this decline is probably further exacerbated in nature where the food resources are ephemeral. Therefore, small differences in fitness of two-day-old worms may be largely invisible to selection. However, while the decline in selection gradients with age is particularly strong in *C. elegans*, the reduced force of selection with advancing age is a general pattern across organisms.

Our findings support the hypothesis that gene expression is optimized for development and early-life reproduction

across a broad range of physiological processes. Consequently, gene expression in adulthood can be optimized further to improve survival and, potentially, fitness. Only *ife-2* adulthood-only RNAi animals had simultaneously increased survival and egg size; however, longevity usually correlates with increased resistance to different ecologically relevant stressors, such as temperature, light and pathogens [64,65], so it is possible that improved survival could contribute positively to fitness under more challenging conditions in nature. Moreover, there is little evidence that the fitness cost is in the next generation. The trade-off between number and quality of offspring is well known in the life-history literature [66], and in *C. elegans* reduced egg size results in impaired offspring performance [67]. We find, however, no evidence for reduced egg size in any treatment; instead, in the few cases when egg size was affected the RNAi treatment resulted in larger eggs. In line with this, adult downregulation of *daf-2* in *C. elegans* increases offspring performance, partly mediated by increased egg size [23]. These results of course do not preclude the possibility that other types of trade-offs contribute to ageing in *C. elegans* [9]. One important aspect to consider is phenotypic plasticity and how these animals would perform in different contexts [8]. Future work should focus on studying fitness consequences of age-specific gene expression optimization across a broad range of ecologically relevant environments.

Data accessibility. Data available at https://doi.org/10.6084/m9.figshare.12666416.v2.

Authors' contributions. A.A.M., H.C. and M.I.L. conceived the study; H.C., E.M.L.D. and E.I.-C. collected the data; M.I.L., E.M.L.D. and E.I.-C. analysed the data; A.A.M. wrote the draft; all authors contributed to writing the final version.

Competing interests. We declare we have no competing interests.

Funding. This work has been supported by BBSRC BB/R017387/1 and ERC Consolidator Grant GermlineAgeingSoma 724909 to A.A.M. and Swedish Research Council grant no. 2016-05195 (Vetenskapsrådet) to M.I.L.

Acknowledgements. We thank Alper Akay and Roberta Skukan for their molecular advice and reagents for the gene expression assay, and Kris Sales for his experimental assistance with the nematode work.

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
