## [Reviewer comments · Proceedings of the Royal Society B: Biological Sciences]

Review History

RSPB-2020-1728.R0 (Original submission)

Review form: Reviewer 1

Recommendation

Major revision is needed (please make suggestions in comments)

Scientific importance: Is the manuscript an original and important contribution to its field?

Good

General interest: Is the paper of sufficient general interest?

Excellent

Quality of the paper: Is the overall quality of the paper suitable?

Good

Is the length of the paper justified?

Yes

Should the paper be seen by a specialist statistical reviewer?

No

Do you have any concerns about statistical analyses in this paper? If so, please specify them explicitly in your report.

No

It is a condition of publication that authors make their supporting data, code and materials available - either as supplementary material or hosted in an external repository. Please rate, if applicable, the supporting data on the following criteria.

Is it accessible?

Yes

Is it clear?

Yes

Is it adequate?

Yes

Do you have any ethical concerns with this paper?

No

Comments to the Author

In this paper the authors used a candidate gene approach to test for the effects of gene expression at different life stages on lifespan and reproductive function in *C. elegans*. The authors indeed detect age-specific effects on lifespan and reproduction, and that the two do not necessarily trade-off with one another. They conclude that in many cases gene expression is not optimised for the adult life stage, most likely as a result of the diminishing power of selection with age.

I would like to highlight that I am not familiar with model system or RNAi technique used so largely contain my review to concepts and analyses.

There is a lot in here for readers from different backgrounds (e.g. physiology, evolutionary, genetics) so I think a general biology journal such as Proc B is an excellent outlet. I found the results interesting and the paper really made me think; I am sure others in the field will find it interesting too. The paper does a good job of highlighting issues with DSA, which I think most in field have already started to abandon. But more than this, it also highlights some interesting ways forward. The analyses are appropriate. The paper is mostly well written, but I think a bit of repackaging particularly around the intro would help the reader get the most out of it.

General Comments

Intro and DTA

My feeling here is that the description and arguments around DTA are missing something. There is a difference between selection being weak, and selection being constrained (e.g. due to a tradeoff - note necessarily a resource-based tradeoff as in DSA). Even if selection is weak at later age classes, should it not still optimise gene expression at those ages (eventually) even if the gains in fitness are marginal? If it is constrained, however, because optimising at early ages makes optimising at later ages impossible, this is different.

Intro and Individual Fitness

I really enjoyed that the authors had integrated their life-histories into individual fitness. It settled all kinds of questions I had about how the potential trade-offs between repro and survival at the different age classes came together. I would add a sentence highlighting this in the final paragraph of the intro.

Methods

The authors do not test that RNAi is having its desired effects on expression in the targeted pathways. Can they at least point to studies that have shown this?

Analyses

Did the authors consider models with time-varying coefficients to explore the effects of the treatments on age-specific mortality? I.e. Is Adult mortality the same in Lifelong RNAi flies as it is Adulthood RNAi flies? This may provide a window into the mechanism. There could be a late-life cost to the early-life RNAi treatment.

Minor

L64: effects not affects.

L69-71: It is often overlooked, but fundamental assumption of this hypothesis is that soma and reproductive function are competing for same pool of resources. I think this should be explicitly stated.

L99: "Evolutionarily conserved".

L211: Highlight that 'Treatment' here is the age of RNAi exposure.

Methods: Consider presenting a covariance matrix of life-history traits.

L306: "The corollary of this argument IS that" – also RE this point, unless optimisation at the adult stage is constrained by optimising at younger classes (i.e. comment above).

L332: 'IN other words, early ...'

Discussion

L347: Yes, this is a curious finding – it suggests that age-1 expression is not at all optimised in this population. Other explanations (beyond the context) are 1) there is an intergenerational cost of the treatment to F1 survival that were not quantified in the experiment; 2) there are costs to developmental time not captured (i.e. a uniform 2 days for calculating fitness is not appropriate).

Table 2: Not clear which coefficients are for which traits.

Review form: Reviewer 2

Recommendation

Accept with minor revision (please list in comments)

Scientific importance: Is the manuscript an original and important contribution to its field?

Excellent

General interest: Is the paper of sufficient general interest?

Good

Quality of the paper: Is the overall quality of the paper suitable?

Excellent

Is the length of the paper justified?

Yes

Should the paper be seen by a specialist statistical reviewer?

No

Do you have any concerns about statistical analyses in this paper? If so, please specify them explicitly in your report.

Yes

It is a condition of publication that authors make their supporting data, code and materials available - either as supplementary material or hosted in an external repository. Please rate, if applicable, the supporting data on the following criteria.

Is it accessible?

Yes

Is it clear?

Yes

Is it adequate?

Yes

Do you have any ethical concerns with this paper?

No

Comments to the Author

Two main theories have been proposed to explain the mechanisms of aging evolution, DST and DTA. This study investigated the effects of gene expression levels of a number of known "aging" genes on lifespan, reproduction and fitness by manipulating them with RNAi in an age-specific manner. This approach is novel in this field and provides interesting and important new insights into the mechanisms of aging. The design of the study and the results are robust, although I still have some questions about the analyses to be clarified. Based on the results, the authors conclude that their results mostly support the DTA. Although I agree with this in general, I have a few comments in this respect. Overall, this MS is well written and much of interest to the field of aging and life history research.

Comments:

1. The title suggests that you observed "cost-free" lifespan extension, which is a premature claim given that this study focused on the correlation of lifespan and reproduction. As the authors mention as well, other trade-offs may exist and fitness effects may be context dependent.
2. Do you have an indication of how much your five genes are downregulated by your RNAi treatment, and how "strong" this manipulation is compared to naturally occurring variation in gene expression of these genes? If your knockdown is much stronger than what would occur in nature, then your results and their interpretation (also with regard to DTA/DST) may not necessarily be representative of the processes that control natural variation in aging.
3. P6+7, L107-128: I am missing a description of the gene *igf-1* and what is known of this gene with regard to aging and reproduction.
4. P8L152: "recovered from frozen" -- do you mean "frozen stocks"?
5. P9L188 and P10L196: "giving x worms per gene" -- you mean xx worms per gene/treatment combination, e.g. 30 worms for each of the three treatments in case of the reproductive assays, 90 worms in total per gene?
6. What unit was used for egg size and how was it calculated/measured?
7. Has it been shown that egg size is a good proxy for offspring quality in worms?
8. The supplementary files show contrasts of the RNAi treatments against the control, but it seems more relevant to compare the three RNAi treatments amongst each other, or most

importantly to compare the adult treatment(s) to the lifelong treatment, given that your aim is to investigate the effect of age-specific RNAi not the effect of RNAi in general.

9. Why was Age*Age added to the model and how can you interpret a (non-)significant effect for this factor?

Decision letter (RSPB-2020-1728.R0)

09-Sep-2020

Dear Dr Maklakov:

Your manuscript has now been peer reviewed and the reviews have been assessed by an Associate Editor. The reviewers' comments (not including confidential comments to the Editor) and the comments from the Associate Editor are included at the end of this email for your reference. As you will see, the reviewers and the Editors have raised some concerns with your manuscript and we would like to invite you to revise your manuscript to address them.

Research ethics:

Use of animals and field studies:

It is a condition of publication that you make available the data and research materials supporting the results in the article. Please see our Data Sharing Policies (<https://royalsociety.org/journals/authors/author-guidelines/#data>). Datasets should be deposited in an appropriate publicly available repository and details of the associated accession number, link or DOI to the datasets must be included in the Data Accessibility section of the article (<https://royalsociety.org/journals/ethics-policies/data-sharing-mining/>). Reference(s) to datasets should also be included in the reference list of the article with DOIs (where available).

Please submit a copy of your revised paper within three weeks. If we do not hear from you within this time your manuscript will be rejected. If you are unable to meet this deadline please let us know as soon as possible, as we may be able to grant a short extension.

Best wishes,
Dr Locke Rowe
mailto: proceedingsb@royalsociety.org

Associate Editor

Comments to Author:

This is a very interesting study, which tests two competing hypothesis for the evolution of aging - - Disposable Soma vs Developmental Theory of Aging. The latter suggests the aging occurs because gene expression is not optimised for late in life (selection optimises expression of genes for early in life). Here, Maklakov et al use RNAi to reduce levels of gene expression a 5 key "aging" genes in *C. elegans* at 3 different ontogenetic stages, showing that downregulation during or after reproductive age can increase longevity, in many case without incurring cost to reproductive success. This provides evidence for the Developmental Theory of Aging. It is a neat

paper, and was sent to peer review by 2 referees. Each found the paper to be interesting, and each has raised a number of insightful comments, all of which require the authors to address.

Referee 1 points out that some repackaging of the conceptual background in the Introduction, with respect to the DTA. The referee suggest the authors consider models with time varying coefficients. They also wonder whether the RNAi actually works, since the authors are curiously silent on validating this approach, or cross referencing of other papers from their lab to show the RNAi was effective.

Referee 2 also provides a series of very insightful comments. For example, they also wonder about the RNAi treatment, and also ask whether the extent of downregulation is ecologically relevant (to levels that compare to natural levels of genetic variation in the worms).

In my initial reading, independently of the referees, I had flagged that I wanted to see some convincing evidence (gene expression data of the targeted genes) or more explicit cross-referencing to studies conducted by this lab group that shows the RNAi treatments work as intended (since everything in the study depends on this).

Reviewer(s)' Comments to Author:

Referee: 1

Comments to the Author(s)

In this paper the authors used a candidate gene approach to test for the effects of gene expression at different life stages on lifespan and reproductive function in *C. elegans*. The authors indeed detect age-specific effects on lifespan and reproduction, and that the two do not necessarily trade-off with one another. They conclude that in many cases gene expression is not optimised for the adult life stage, most likely as a result of the diminishing power of selection with age.

I would like to highlight that I am not familiar with model system or RNAi technique used so largely contain my review to concepts and analyses.

There is a lot in here for readers from different backgrounds (e.g. physiology, evolutionary, genetics) so I think a general biology journal such as Proc B is an excellent outlet. I found the results interesting and the paper really made me think; I am sure others in the field will find it interesting too. The paper does a good job of highlighting issues with DSA, which I think most in field have already started to abandon. But more than this, it also highlights some interesting ways forward. The analyses are appropriate. The paper is mostly well written, but I think a bit of repackaging particularly around the intro would help the reader get the most out of it.

General Comments

Intro and DTA

My feeling here is that the description and arguments around DTA are missing something. There is a difference between selection being weak, and selection being constrained (e.g. due to a tradeoff - note necessarily a resource-based tradeoff as in DSA). Even if selection is weak at later age classes, should it not still optimise gene expression at those ages (eventually) even if the gains in fitness are marginal? If it is constrained, however, because optimising at early ages makes optimising at later ages impossible, this is different.

Intro and Individual Fitness

I really enjoyed that the authors had integrated their life-histories into individual fitness. It settled all kinds of questions I had about how the potential trade-offs between repro and survival at the different age classes came together. I would add a sentence highlighting this in the final paragraph of the intro.

Methods

The authors do not test that RNAi is having its desired effects on expression in the targeted pathways. Can they at least point to studies that have shown this?

Analyses

Did the authors consider models with time-varying coefficients to explore the effects of the treatments on age-specific mortality? I.e. Is Adult mortality the same in Lifelong RNAi flies as it is Adulthood RNAi flies? This may provide a window into the mechanism. There could be a late-life cost to the early-life RNAi treatment.

Minor

L64: effects not affects.

L69-71: It is often overlooked, but fundamental assumption of this hypothesis is that soma and reproductive function are competing for same pool of resources. I think this should be explicitly stated.

L99: "Evolutionarily conserved".

L211: Highlight that 'Treatment' here is the age of RNAi exposure.

Methods: Consider presenting a covariance matrix of life-history traits.

L306: "The corollary of this argument IS that" – also RE this point, unless optimisation at the adult stage is constrained by optimising at younger classes (i.e. comment above).

L332: 'IN other words, early ...'

Discussion

L347: Yes, this is a curious finding – it suggests that age-1 expression is not at all optimised in this population. Other explanations (beyond the context) are 1) there is an intergenerational cost of the treatment to F1 survival that were not quantified in the experiment; 2) there are costs to developmental time not captured (i.e. a uniform 2 days for calculating fitness is not appropriate).

Table 2: Not clear which coefficients are for which traits.

Referee: 2

Comments to the Author(s)

Two main theories have been proposed to explain the mechanisms of aging evolution, DST and DTA. This study investigated the effects of gene expression levels of a number of known "aging" genes on lifespan, reproduction and fitness by manipulating them with RNAi in an age-specific manner. This approach is novel in this field and provides interesting and important new insights into the mechanisms of aging. The design of the study and the results are robust, although I still have some questions about the analyses to be clarified. Based on the results, the authors conclude that their results mostly support the DTA. Although I agree with this in general, I have a few comments in this respect. Overall, this MS is well written and much of interest to the field of aging and life history research.

Comments:

1. The title suggests that you observed "cost-free" lifespan extension, which is a premature claim given that this study focused on the correlation of lifespan and reproduction. As the authors mention as well, other trade-offs may exist and fitness effects may be context dependent.

2. Do you have an indication of how much your five genes are downregulated by your RNAi treatment, and how "strong" this manipulation is compared to naturally occurring variation in gene expression of these genes? If your knockdown is much stronger than what would occur in nature, then your results and their interpretation (also with regard to DTA/DST) may not necessarily be representative of the processes that control natural variation in aging.
3. P6+7, L107-128: I am missing a description of the gene *igf-1* and what is known of this gene with regard to aging and reproduction.
4. P8L152: "recovered from frozen" -- do you mean "frozen stocks"?
5. P9L188 and P10L196: "giving x worms per gene" -- you mean xx worms per gene/treatment combination, e.g. 30 worms for each of the three treatments in case of the reproductive assays, 90 worms in total per gene?
6. What unit was used for egg size and how was it calculated/measured?
7. Has it been shown that egg size is a good proxy for offspring quality in worms?
8. The supplementary files show contrasts of the RNAi treatments against the control, but it seems more relevant to compare the three RNAi treatments amongst each other, or most importantly to compare the adult treatment(s) to the lifelong treatment, given that your aim is to investigate the effect of age-specific RNAi not the effect of RNAi in general.
9. Why was Age*Age added to the model and how can you interpret a (non-)significant effect for this factor?

Author's Response to Decision Letter for (RSPB-2020-1728.R0)

See Appendix A.

RSPB-2020-1728.R1 (Revision)

Review form: Reviewer 1

Recommendation

Accept as is

Scientific importance: Is the manuscript an original and important contribution to its field?

Excellent

General interest: Is the paper of sufficient general interest?

Excellent

Quality of the paper: Is the overall quality of the paper suitable?

Excellent

Is the length of the paper justified?

Yes

Should the paper be seen by a specialist statistical reviewer?

No

Do you have any concerns about statistical analyses in this paper? If so, please specify them explicitly in your report.

No

It is a condition of publication that authors make their supporting data, code and materials available - either as supplementary material or hosted in an external repository. Please rate, if applicable, the supporting data on the following criteria.

Is it accessible?

Yes

Is it clear?

Yes

Is it adequate?

Yes

Do you have any ethical concerns with this paper?

No

Comments to the Author

The authors have addressed all of my concerns satisfactorily. I recommend publication.

Decision letter (RSPB-2020-1728.R1)

11-Jan-2021

Dear Dr Maklakov

I am pleased to inform you that your manuscript entitled "Cost-free lifespan extension via optimisation of gene expression in adulthood aligns with the developmental theory of ageing" has been accepted for publication in Proceedings B.

Open Access

Paper charges

Sincerely,
Dr Locke Rowe
Editor, Proceedings B
mailto: proceedingsb@royalsociety.org

Associate Editor:
Board Member: 1
Comments to Author:

The authors have done a superb job with the revision, and I believe the paper will be of great interest to many readers of Proceedings B. The paper was sent back to one of the referees, who was completely satisfied with the new version. I thank the authors for their efforts to validate the RNAi expression data, and the extra age-specific mortality rate analyses. On my own part, I apologise for the delay in getting to this paper - it coincided with a period of annual leave in which I was away from the computer. Please see a few comments that require correction / clarification in the final paper, below.

Line 130 – Description of nuo-6 is confusing, since as written it could be interpreted that the gene is encoded by mtDNA (the authors use the phrase ‘mitochondrial subunit of Complex 1’); but I believe the gene is probably a nuclear-encoded subunit of mitochondrial complex 1. Can the authors clarify.

Table S1 – typo in final sentence of caption. Should read “The numbers are a multiple of 10.....

Statistical analyses: Are results using proportional hazards models better defined as “survival” rather than “lifespan”?

Line 265 – “increases over time follow a power function”

Line 274 – Define “Age” here – although it is rather intuitive, it might cause some confusion since it is not explicitly defined. It is the day of measurement of reproductive success.

Line 652 – reference 50 is currently all in capitals.

Board Member: 2
Comments to Author:
(There are no comments.)

Board Member: 3
Comments to Author:
(There are no comments.)

Appendix A

Response to Reviewers

Associate Editor

Comments to Author:

This is a very interesting study, which tests two competing hypothesis for the evolution of aging -- Disposable Soma vs Developmental Theory of Aging. The latter suggests the aging occurs because gene expression is not optimised for late in life (selection optimises expression of genes for early in life). Here, Maklakov et al use RNAi to reduce levels of gene expression a 5 key "aging" genes in *C. elegans* at 3 different ontogenetic stages, showing that downregulation during or after reproductive age can increase longevity, in many case without incurring cost to reproductive success. This provides evidence for the Developmental Theory of Aging. It is a neat paper, and was sent to peer review by 2 referees. Each found the paper to be interesting, and each has raised a number of insightful comments, all of which require the authors to address.

REPLY: Thank you for such a positive evaluation of our manuscript!

Referee 1 points out that some repackaging of the conceptual background in the Introduction, with respect to the DTA. The referee suggest the authors consider models with time varying coefficients. They also wonder whether the RNAi actually works, since the authors are curiously silent on validating this approach, or cross referencing of other papers from their lab to show the RNAi was effective.

REPLY: We discuss the conceptual points in detail in response to Reviewer 1. We run additional models using BASTA. We also present rt-qPCR results quantifying reduction in gene expression. We note, however, that we used standard clones for five extremely well described and well-studied genes and recovered all classical lifespan phenotypes for all genes. In other words, we reproduced lifespan extension that was found in the previous studies by other labs using the same RNAi approach.

Referee 2 also provides a series of very insightful comments. For example, they also wonder about the RNAi treatment, and also ask whether the extent of downregulation is ecologically relevant (to levels that compare to natural levels of genetic variation in the worms).

REPLY: This is a very interesting and important comment. While it is unknown what represents an ecologically relevant age-specific gene expression in *C. elegans*, we explain that testing the predictions derived from DTA and DST theories does not require mimicking natural gene expression. This is because antagonistically pleiotropic alleles, which are beneficial for fitness, are often expected to go to fixation. Our approach does not rely on standing genetic variation for age-specific expression.

Reviewer(s)' Comments to Author:

Referee: 1

Comments to the Author(s)

In this paper the authors used a candidate gene approach to test for the effects of gene expression at different life stages on lifespan and reproductive function in *C. elegans*. The authors indeed detect age-specific effects on lifespan and reproduction, and that the two do not necessarily trade-off with one another. They conclude that in many cases gene expression is not optimised for the adult life stage, most likely as a result of the diminishing power of selection with age.

I would like to highlight that I am not familiar with model system or RNAi technique used so largely contain my review to concepts and analyses.

There is a lot in here for readers from different backgrounds (e.g. physiology, evolutionary, genetics) so I think a general biology journal such as Proc B is an excellent outlet. I found the results interesting and the paper really made me think; I am sure others in the field will find it interesting too. The paper does a good job of highlighting issues with DSA, which I think most in field have already started to abandon. But more than this, it also highlights some interesting ways forward. The analyses are appropriate. The paper is mostly well written, but I think a bit of repackaging particularly around the intro would help the reader get the most out of it.

REPLY: We thank the reviewer for the positive view of the paper.

General Comments

Intro and DTA

My feeling here is that the description and arguments around DTA are missing something. There is a difference between selection being weak, and selection being constrained (e.g. due to a tradeoff – note necessarily a resource-based tradeoff as in DSA). Even if selection is weak at later age classes, should it not still optimise gene expression at those ages (eventually) even if the gains in fitness are marginal? If it is constrained, however, because optimising at early ages makes optimising at later ages impossible, this is different.

REPLY: This is an extremely interesting and important point to discuss. Weakened selection (a selection shadow) after sexual maturity is in the heart of all evolutionary theories of ageing, and for DTA, this selection shadow is the ultimate cause why gene expression is predicted to be optimized for development and not for late life.

Essentially, both DTA and DSA stem directly from Williams 1957 antagonistic pleiotropy theory. The key point is that selection on positive early-life effect is strong, while selection against detrimental late-life effect is weak. This model does not imply that optimization is impossible, just that selection in late-life is not strong enough to

optimize it. We discuss this in detail in a recent review (Maklakov and Chapman 2019, Proc B; see also de Magalhaes and Church 2005 for the introduction to DTA). In fact, the proximate DTA theory is the most direct case of antagonistic pleiotropy as envisioned by Williams in 1957 paper.

Theory suggests that we should not expect selection to optimize fitness in late ages. Moreover, the second major population genetic theory of ageing, Medawar's mutation accumulation (MA), rests on the idea that selection cannot optimize late-life fitness even in the absence of early-life benefit. Since there is no theoretical objection for the evolution of ageing via mutation accumulation, there is even less theoretical objection for the evolution of ageing via antagonistic pleiotropy.

Intro and Individual Fitness

I really enjoyed that the authors had integrated their life-histories into individual fitness. It settled all kinds of questions I had about how the potential trade-offs between repro and survival at the different age classes came together. I would add a sentence highlighting this in the final paragraph of the intro.

REPLY: We are happy that you see the benefit of using a composite fitness measure, and we have followed the suggestion to further motivate the use of this measure of fitness in the last paragraph of the introduction.

Methods

The authors do not test that RNAi is having its desired effects on expression in the targeted pathways. Can they at least point to studies that have shown this?

REPLY: We have now validated the downregulation of gene expression for our targeted genes, following feeding RNAi. We performed qRT-PCR and present the results that demonstrate a fold change reduction in two day adults exposed to the relevant RNAi treatment from egg, compared with expression of these respective genes in untreated age-matched controls (based on gene expression fold change, $2^{-\Delta\Delta CT}$). Please see Methods and Supplementary materials, in particular Table S21 and Figures S7 and S8.

We also note that we recover all the classical phenotypes with respect to lifespan extension in all the genes that we tested.

Analyses

Did the authors consider models with time-varying coefficients to explore the effects of the treatments on age-specific mortality? I.e. Is Adult mortality the same in Lifelong RNAi flies as it is Adulthood RNAi flies? This may provide a window into the mechanism. There could be a late-life cost to the early-life RNAi treatment.

REPLY: While DTS and DTA makes no explicit predictions about age-specific mortality rate, we have now analysed mortality rate using the package BaSTA, where we have fitted a either a Gompertz, Weibull or

logistic model with a simple, Makeham or bathtub shape and for each gene performed model selection to arrive at the best model, which has been evaluated (see Methods for details). The results of these models are now available as supplementary figures S1-S5 and supplementary tables S4-S13. We also discuss the results at lines 493-500.

While different mortality rate models were suitable for different genes, we found that lifespan extension was achieved by decreased age-specific mortality rate and in some RNAi treatments also by lowered baseline mortality. However, for *raga-1*, down-regulation by RNAi instead resulted in a lowered late-life asymptote in mortality rate, again suggesting that the two major nutrient signalling pathways IIS (*age-1*) and TOR (*raga-1*) have different effects on life-history traits. Differences between RNAi treatments were sometimes caused by changed baseline mortality (*age-1*, *raga-1*, *nuo-6*), sometimes by altered age-specific mortality rate, (*nuo-6*, *ifg-1*, *ife-2*) and in one case also by changes in late-life asymptote (*ifg-1*).

Minor

L64: effects not affects.

REPLY: Corrected.

L69-71: It is often overlooked, but fundamental assumption of this hypothesis is that soma and reproductive function are competing for same pool of resources. I think this should be explicitly stated.

REPLY: We agree, and this is now explicitly stated in the sentence, see lines 71-72.

L99: "Evolutionarily conserved".

REPLY: Corrected.

L211: Highlight that 'Treatment' here is the age of RNAi exposure.

REPLY: The suggestion is now added for clarification.

Methods: Consider presenting a covariance matrix of life-history traits.

REPLY: Since all traits are not measured from the same individuals (especially, lifespan stems from a separate experiments) we do not find it ideal to perform an analysis of phenotypic covariances. Moreover, given the large number of cells in such a matrix (3 trait covariances, for each of 4 treatment levels per gene) we would needed a different experimental protocol to estimate all combinations with accuracy. Therefore, while the idea is interesting, we do not find such an analysis suitable for the current study.

L306: "The corollary of this argument IS that"

REPLY: Corrected.

– also RE this point, unless optimisation at the adult stage is constrained by optimising at younger classes (i.e. comment above).

REPLY: Yes, indeed, this is the premise of the DTA that optimization at adult stage is constrained by fitness effects at younger classes.

However, we can experimentally break-up this constrain by downregulating the expression at ages of interest. Presumably, selection on the evolution of such a modifier gene that optimizes the expression of our genes of interest in late-life is not sufficiently strong. The question is whether we should then call it a constraint at all. This is an interesting and important discussion that is probably outside the scope of this paper.

L332: 'IN other words, early ...'

REPLY: The sentence is now corrected.

Discussion

L347: Yes, this is a curious finding – it suggests that age-1 expression is not at all optimised in this population. Other explanations (beyond the context) are 1) there is an intergenerational cost of the treatment to F1 survival that were not quantified in the experiment; 2) there are costs to developmental time not captured (i.e. a uniform 2 days for calculating fitness is not appropriate).

REPLY: These are possible explanations, and we have approached them in the following way:

For suggestion (1), we have measured egg size, which is strongly correlated to offspring performance (Perez et al. 2017 Nature) in order to detect a possible intergenerational cost. We do not find any evidence of parents trading off size and number of eggs. Even though other intergenerational costs may exist, the major cost is controlled for. This is now discussed at lines 525-531.

For suggestion (2) this is actually taken into account, but we see that it was not explained properly. The added two extra days are the days of juvenile development (with no reproduction), egg collection start on day 3, when worms are maturing and starting to lay eggs, and continue for their whole reproductive period. If a worm is developing quickly (and reproduction is kept constant), more of its reproduction will take place during the first day of collection (day 3) since all worms spend part of this day as late-stage juveniles maturing into adults at different times. Therefore, the fitness measure will specifically take fast development (and therefore early reproduction) into account. This is now explained in the methodology, lines 297-298.

In summary, while other costs may exist (please see last section of discussion), we have largely controlled for the two potential costs outlined above.

Table 2: Not clear which coefficients are for which traits.

REPLY: The table in the previous submission was, by mistake, duplicated. We have now fixed this mistake. Therefore, all coefficients are for the trait Age-specific reproduction.

Referee: 2

Comments to the Author(s)

Two main theories have been proposed to explain the mechanisms of aging evolution, DST and DTA. This study investigated the effects of gene expression levels of a number of known "aging" genes on lifespan, reproduction and fitness by manipulating them with RNAi in an age-specific manner. This approach is novel in this field and provides interesting and important new insights into the mechanisms of aging. The design of the study and the results are robust, although I still have some questions about the analyses to be clarified. Based on the results, the authors conclude that their results mostly support the DTA. Although I agree with this in general, I have a few comments in this respect. Overall, this MS is well written and much of interest to the field of aging and life history research.

Comments:

1. The title suggests that you observed "cost-free" lifespan extension, which is a premature claim given that this study focused on the correlation of lifespan and reproduction. As the authors mention as well, other trade-offs may exist and fitness effects may be context dependent.

REPLY: We fully agree with the Reviewer that fitness effects may be context dependent. In fact, we are currently developing a research program focusing on context-dependence of age-specific gene function in the context of lifespan and ageing. We therefore thought carefully about the possibility of changing the title in accordance with the Reviewer's suggestion. However, we decided that the current title reflects the findings in the environment in which this experiment was performed. We did estimate individual fitness and we even looked at egg size as a proxy for offspring quality. Therefore, we feel that we have a good grasp of fitness in this environment. Most importantly, we believe that the theories that we test here (DST and DTA) provide unique prediction that can be tested in this environment, as DTA for example focuses on the trade-off between survival, growth and reproduction. It will be instructive, although logistically challenging, to study this across a range of different environments in the future.

2. Do you have an indication of how much your five genes are downregulated by your RNAi treatment, and how "strong" this manipulation is compared to naturally occurring variation in gene expression of these genes? If your knockdown is much stronger than what would occur in nature, then your results and their interpretation (also with regard to DTA/DST) may not necessarily be representative of the processes that control natural variation in aging.

REPLY: This is a very interesting point to consider. We now present the results of gene expression downregulation (please see above). However, we believe that it does not relate to the question at hand.

Crucially, we did not intend to downregulate gene expression in way that mimics the natural variation. It is important to note, that studying natural variation in gene expression is not always the best way to

separate between these two theories (DST and DTA). In fact, it may be not possible to all to do so by focusing on natural variation. While we do not know that exact contribution of different alleles (i.e. MA and AP alleles) to ageing in the wild, it is quite possible that many AP alleles go to fixation in natural populations, in particular AP alleles with large effect size. Remember that these are beneficial alleles that will be selected for. Thus, such alleles may not be necessarily contributing to natural variation in lifespan and ageing. Nevertheless, such alleles could be the ones defining lifespan in a broad sense, while the remaining standing genetic variation could be explained mostly by small-effect alleles maintained by mutation-selection balance.

The rationale behind our approach here is to test the theoretical possibility of age-specific gene expression that extends lifespan without negative consequences for development/reproduction. We did not intend to test whether this represents natural variation in gene expression in *C. elegans*.

3. P6+7, L107-128: I am missing a description of the gene *igf-1* and what is known of this gene with regard to aging and reproduction.

REPLY: This information was omitted by mistake, and is now present in the introduction (lines 121-124). Many thanks for spotting this mistake. The *ifg-1* encodes the *C. elegans* orthologue of the scaffold protein eIF4G, a part of the eIF4F complex, which mediates mRNA translation. Inhibition of *ifg-1* increases lifespan but reduces fecundity and slows down growth.

4. P8L152: "recovered from frozen" -- do you mean "frozen stocks"?

REPLY: Yes, "stocks" has now been added to the sentence.

5. P9L188 and P10L196: "giving x worms per gene" -- you mean xx worms per gene/treatment combination, e.g. 30 worms for each of the three treatments in case of the reproductive assays, 90 worms in total per gene?

REPLY: Yes, we mean per gene and treatment combination, this has now been clarified in the manuscript. See lines 196 and 205. Since it is four treatment combinations, we use in total $4 \times 30 = 120$ worms per gene for reproduction, and $4 \times 100 = 400$ worms per gene for lifespan.

6. What unit was used for egg size and how was it calculated/measured?

REPLY: Egg size was measured from photos as cross-section area, in mm^2 . This is now added to the methodology.

7. Has it been shown that egg size is a good proxy for offspring quality in worms?

REPLY: Yes, the size of the egg positively affects offspring life-history traits such as fecundity and development (Perez et al. 2017, Nature). Moreover, *daf-2* down-regulation in adulthood results in increased offspring performance, partly resulting from an increased egg size (Lind et al. 2019. *Evol Lett.*). This is now discussed at lines 525-530.

8. The supplementary files show contrasts of the RNAi treatments against the control, but it seems more relevant to compare the three RNAi treatments amongst each other, or most importantly to compare the adult treatment(s) to the lifelong treatment, given that your aim is to investigate the effect of age-specific RNAi not the effect of RNAi in general.

REPLY: Our focus is on lifespan extension, where a significant lifespan extension (vs control) but lack of a significant fitness cost (vs control) is crucial tests to distinguish DST from DTA. In contrast, whether the adult and lifelong treatments differ can not be used to distinguish these theories, since it is the effect vs control that matters. Therefore, we choose to keep our current a priori analytical approach.

9. Why was Age*Age added to the model and how can you interpret a (non-)significant effect for this factor?

REPLY: Since age-specific reproduction is bell-shaped (increasing to a peak at day 2, then decreasing), Age² was added to models of age-specific reproduction to capture the curvature. The results are presented in table 3 and shows that the curvature is always present and significant. A non-significant effect of Age² would, if present, imply that there is no significant curvature. That is however not the case.

We have now explaining that Age² is in the model to capture curvature, see line 275.

One should however not mistake the Age² term in table S18 for the curvature of reproduction. This term is instead allowing the over/under-dispersion of the model (if present) vary with the level of the covariate (that is, with Age and Age²).